# OpenReview forum: "From Noise to Laws: Regularized Time-Series Forecasting via Denoised Dynamic Graphs"
_ICLR.cc/2026/Conference — ICLR 2026 Conference Withdrawn Submission_

### Official Review · Reviewer_Tr8a · 2025-10-30

**Soundness:** 2
**Presentation:** 1
**Contribution:** 1
**Rating:** 2
**Confidence:** 2

**Summary:**

The authors have proposed a model for long-horizon multivariate time series forecasting, called PRISM.

The architecture consists of:
- a score-based diffusion pre-conditioner
- graph encoder
- forecast head
- physics-based penalties

**Strengths:**

The authors have followed a good system of presenting results, trying to explain them, and then perform ablation studies to establish that each block in the architecture is useful.

**Weaknesses:**

1. The writing is dense, and the paper overall is difficult to follow
2. There is heavy use of technical jargon, from the abstracts to the very end, the conclusion.
3. Numerical results are presented without confidence intervals
4. The resolution of Fig 1 is low, and the latex equations are not rendered correctly
5. Some equations are numbered, while others are not.

**Questions:**

1. What is the rationale behind averaging the performance on different prediction lengths?
2. Can the authors please report the 95% confidence intervals for the results, esp. the ablation studies, to know that the performance drops are statistically significant?
3. Why is the maximum number of neighbours called $k_{\min}$?
4. What does the performance gain trend look like across the dataset size and number of training samples?
5. Have the authors performed any time-profiling of PRISM in comparison to the baselines?
6. Could the authors please provide a TL;DR version of the work without using any technical jargon?

---

### Official Review · Reviewer_MufC · 2025-10-30

**Soundness:** 2
**Presentation:** 1
**Contribution:** 2
**Rating:** 2
**Confidence:** 4

**Summary:**

The paper proposes a long-horizon multivariate forecasting pipeline that combines pre-conditions inputs with a diffusion-style denoiser, builds correlation-thresholded dynamic graphs for message passing, and applies structure/physics-inspired regularizers (range/kinematics, PDE residual) to stabilize rollouts. Claims include improved stability and accuracy on common LTSF benchmarks. The paper misses out on mentioning multiple related works. Overall the paper presents a sensible recipe (denoising + dynamic graphs + physics-inspired regularizers) but the paper reads hand-wavy, exact novelty (beyond stitching different components together) is unclear and references and experiments are severely lacking. I recommend Reject pending a thorough revision and stronger experimental comparisons.

**Strengths:**

Following are the strengths I can identify :

Coherent, practical recipe. Combining input denoising, dynamic graph structure, and lightweight physical constraints is a reasonable direction for stabilizing long-horizon predictions.

Some theoretical intuitions are outlined (contraction/conditioning arguments) that aim to justify stability.

Ablation sketch suggests each component helps.

**Weaknesses:**

**Questionable novelty**  : Each module (diffusion denoising, correlation-based dynamic graphs, physics-style penalties) is known; the paper doesn’t clearly isolate what is new beyond the combination, nor why the combination is more than “module stitching.” Although the authors do make a note at the end of the related work section but never explain how everything blends in together.

**Hand-wavy Method section** : The method section seemed hand wavy in terms of defining notations properly. Multiple equations don't have any equation number to refer to e.g. 3.1, 3.2, 3.3 etc. \gamma for example was never defined in 3.3. Notation errors, undefined variables, and ambiguous procedures make the method hard to follow mathematically what's really happening.

**Lacking experiment section** : Needs a broader survey of dynamic graph learning for forecasting, simple and robust linear baselines (DLinear, N-BEATS, N-HiTS), graph forecasters (DCRNN, GraphWaveNet), koopman based methods and classical filters (ARIMA/ETS/Kalman). For denoising/preconditioning, compare against simpler frequency- or state-space methods before adopting diffusion.

**Extremely limited references** : Related work omits several pertinent lines (dynamic graph forecasters, classical/linear baselines, denoising alternatives like AR/Kalman/spectral methods), making it hard to contextualize design choices. Minor point : Multiple duplicates in the reference section.
The paper fails to really use the theoretical results. A discussion of when the theoretical results actually stand vs fail cases would greatly improve the paper.

Following are more direct action items :

**Clarity & Notation needed to be fixed (high priority)**  :

Naming flip/inconsistency. The paper alternates identifiers for the method in figures vs. text.

Shape/notation errors. Examples include swapping
D/
H/
N in window and output shapes.

Undefined variables/ops. Symbols like the correlation-power
γ, reweighting thresholds, “
ϕ-lift,” and graph degree-capping procedures appear without formal definitions.

Train vs. inference ambiguity. It’s unclear whether training uses one-shot or iterative rollout and how exposure bias is handled.

**Soundness of Experiments **  :

Missing variability. All key tables should report mean ± std over ≥3 seeds; current single-number reporting is insufficient for ICLR.

Baseline coverage. Include strong, modern linear and graph baselines; show per-horizon results, not only averages.

Ablations & sensitivity. Vary correlation threshold  τ, degree caps, window  W, and regularizer weights; analyze signed vs. absolute correlations.

**Questions:**

See weaknesses. The paper is far from a polished submission that actually makes a clear contribution to the field.

---

### Official Review · Reviewer_oaug · 2025-11-01

**Soundness:** 2
**Presentation:** 2
**Contribution:** 2
**Rating:** 2
**Confidence:** 4

**Summary:**

The authors propose PRISM, a long-term time series forecasting model combining (1) a score-based diffusion preconditioner for denoising input history, (2) dynamic correlation-thresholded graph encoder with time-varying adjacency from sliding-window correlations, and (3) physics-informed regularizers including range envelopes, velocity/acceleration caps, reaction-diffusion residual. The total loss comprises six terms with five hyperparameters. The paper reports state-of-the-art results on Electricity, Traffic, Weather, ILI, Exchange, and ETT benchmarks.

**Strengths:**

The combination of diffusion-based denoising with dynamic graph encoding and regularization appears to be novel in application to LTSF, even though individual components are established.  The frequency domain analysis showing preserved fundamentals with attenuated high-frequency noise provides useful diagnostic insight. The authors present ablation studies to disentangle contributions of different components.

**Weaknesses:**

I find the "physics-informed" terminology somewhat misleading. The regularizers here are smoothness constraints: velocity/acceleration caps from empirical training percentiles, range clipping from min/max, and a reaction-diffusion residual that is simply graph Laplacian smoothing with learnable parameters. These are heuristic regularizers, not domain physics and invites confusion with genuine physics-based modeling applications where known governing equations are leveraged for data-driven learning.

The theoretical results do not provide meaningful insights. Proposition 1 states "if the spectral radius is less than 1, then it's a contraction"—this is definitional. Proposition 2 is standard for affine compositions. Neither result provides approximation error bounds, or explains when/why the regularizers help.

The six-term loss function with 5+ hyperparameters, plus additional hyperparameters (e.g., $\tau_{threshold}$, $k_{min}$, $W$, diffusion steps), creates substantial risk of overfitting to benchmarks. The ablation states "none dominates across datasets or metrics...the full design is the most consistently strong"—suggesting that only the complete combination of all hyperparameters yields gains.

I did not see any no error bars, confidence intervals, or indication of multiple runs with different seeds. On heavily benchmarked datasets, marginal improvements without significance testing are insufficient to claim superiority.

The diffusion preconditioner lacks analysis of signal-versus-noise tradeoffs or how denoising interacts with downstream loss terms. What signals are preserved? What's removed? How does this affect forecasting accuracy versus a simple low-pass filter?

**Questions:**

- I was unable to find details of the hyperparameter tuning procedure —how were these 10+ hyperparameters selected? Were the baseline methods given equivalent comptutational budgets for tuning?
- Can you provide results with the original evaluation protocol (longer horizons) on ILI and Exchange datasets?
- Report confidence intervals across multiple random seeds for all benchmarks.
- The ablation table shows "none dominates...except full model." Can you identify which subsets of regularizers provide consistent gains across datasets?
- What is the computational cost of the diffusion preconditioner? Can you please provide wall-clock time comparisons including preprocessing.
- For the diffusion denoiser: what signals are preserved versus removed? How does this compare to simple baselines like moving average filters?

---

### Official Review · Reviewer_LhAs · 2025-11-01

**Soundness:** 2
**Presentation:** 1
**Contribution:** 2
**Rating:** 2
**Confidence:** 3

**Summary:**

The paper introduces PRISM, a physics-informed spatiotemporal model for epidemic forecasting that integrates diffusion-based priors and graph neural networks. It provides theoretical stability guarantees through reaction-diffusion analysis and Lipschitz bounds.

**Strengths:**

I appreciate the incorporation of the reaction–diffusion prior on the forecast trajectory and the accompanying theoretical development that establishes its stability guarantees. In this regard, one of the strengths of the paper lies in Propositions 1 and 2, which are commendable for the authors’ effort to provide theoretical grounding. These propositions rely on standard arguments that have been previously established in the graph-stability literature, such as [1].

[1] Gama, F., Ribeiro, A. and Bruna, J., 2020, May. Stability of graph neural networks to relative perturbations. In ICASSP 2020-2020 IEEE International Conference on Acoustics, Speech and Signal Processing (ICASSP) (pp. 9070-9074). IEEE.

**Weaknesses:**

I consider the incorporation of the reaction–diffusion prior to be the only element of genuine novelty in this paper. The remaining components appear to be a compilation of previously developed ideas, assembled in a way that does not provide new conceptual insight or a clear methodological rationale for their integration. The paper lacks an overarching research hypothesis or unifying theoretical perspective; rather, it combines several existing modules with limited justification for their joint use. The emphasis on the novelty of the physics-informed component is also not well founded, as similar approaches have already been explored in prior work (e.g., [2]). The idea of range constraints, while useful, is not novel either.

I would encourage authors to further explore the advantages of the reaction–diffusion prior in other autoregressive deep learning models for time series prediction. I think that would be a promising paper to write. The current version of the paper does not focus much of its attention on examining this part, but instead addresses the overall architecture.

I would also recommend improving the quality of the figures and the writing style of the paper. Also, the paper should explain in more detail what the notation used in propositions 1 and 2 means, and how to make sense of these results. The provided summary is too high-level; it should guide the reader through the meaning of these bounds and their connection to stability. Similarly, for equation (6), which is hard to decipher how that was derived and what is means.

[2] Huang, J., Yang, G., Wang, Z. and Park, J.J., 2024. DiffusionPDE: Generative PDE-solving under partial observation. Advances in Neural Information Processing Systems, 37, pp.130291-130323.

**Questions:**

There are DORIC and PRISM – are they are the same thing?

---

### Note · Authors · 2026-01-10

I have read and agree with the venue's withdrawal policy on behalf of myself and my co-authors.